# Membrane Fluidization Governs the Coordinated Heat-Inducible Expression of Nucleus- and Plastid Genome-Encoded *Heat Shock Protein* 70 Genes in the Marine Red Alga *Neopyropia yezoensis*

**DOI:** 10.3390/plants12112070

**Published:** 2023-05-23

**Authors:** Koji Mikami, Ho Viet Khoa

**Affiliations:** 1School of Food Industrial Sciences, Miyagi University, Hatatate 2-2-1, Sendai 982-0215, Japan; 2Graduate School of Fisheries Sciences, Hokkaido University, 3-1-1 Minato-Cho, Hakodate 041-8611, Japan; hvkhoa59@gmail.com

**Keywords:** heat shock protein 70, heat stress, gene expression, membrane fluidization, *Neopyropia yezoensis*

## Abstract

Heat shock protein 70 (HSP70) is an evolutionarily conserved protein chaperone in prokaryotic and eukaryotic organisms. This family is involved in the maintenance of physiological homeostasis by ensuring the proper folding and refolding of proteins. The HSP70 family in terrestrial plants can be divided into cytoplasm, endoplasmic reticulum (ER)-, mitochondrion (MT)-, and chloroplast (CP)-localized HSP70 subfamilies. In the marine red alga *Neopyropia yezoensis*, the heat-inducible expression of two cytoplasmic *HSP70* genes has been characterized; however, little is known about the presence of other *HSP70* subfamilies and their expression profiles under heat stress conditions. Here, we identified genes encoding one MT and two ER HSP70 proteins and confirmed their heat-inducible expression at 25 °C. In addition, we determined that membrane fluidization directs gene expression for the ER-, MT-, and CP-localized HSP70 proteins as with cytoplasmic HSP70s. The gene for the CP-localized HSP70 is carried by the chloroplast genome; thus, our results indicate that membrane fluidization is a trigger for the coordinated heat-driven induction of *HSP70* genes harbored by the nuclear and plastid genomes in *N. yezoensis*. We propose this mechanism as a unique regulatory system common in the Bangiales, in which the CP-localized HSP70 is usually encoded in the chloroplast genome.

## 1. Introduction

Heat shock proteins (HSPs) were first discovered by the observation of rapid heat-inducible transcription during chromosome puffing in the fruit fly *Drosophila melanogaster* [1,2]. HSPs are ubiquitous, conserved in prokaryotes and eukaryotes, and accumulate under not only heat stress conditions, but also normal conditions and in response to other abiotic and biotic stresses [3,4,5]. The major function of HSPs is the maintenance of protein homeostasis by their chaperone activity, which assists with the folding of three-dimensional proteins during translation and the refolding of denatured proteins under stress conditions [6,7]. Thus, HSPs contribute to the stabilization of proteins, enabling cells to cope with the misfolding, unfolding, degradation, and aggregation of proteins under both optimum and adverse growth conditions.

HSPs are classified into subgroups based on their molecular weight, such as HSP of about 100 kDa (HSP100, also named Clp), HSP90 (HtpG), HSP70 (DnaK), HSP60 (GroEL), HSP40 (DnaJ), HSP10 (GroES), and small HSPs [6,7]. Of these, HSP70 (DnaK) proteins are the most abundant chaperones with high evolutionary conservation [8,9,10], and in collaboration with its co-chaperones HSP40 (DnaJ) and GrpE constitutes a functionally active chaperone [6,11]. Photosynthetic organisms possess four types of HSP70s, which localize to different cellular compartments: the cytoplasm, endoplasmic reticulum (ER), mitochondrion (MT), and chloroplast (CP) [12,13]. Gene duplication is proposed to have occurred following the divergence of the cytoplasm- and ER-localized *HSP70* genes, whereas lateral gene transfer by endosymbiosis established the genes encoding MT- and CP-localized HSP70s in eukaryotic nuclei [14,15]. Based on their distinct localizations, HSP70s play important chaperone roles in a variety of cellular responses.

Much attention has been paid to resolving the transcriptional regulation of *HSP* genes under heat stress conditions. Studies into the underlying signal transduction pathways have identified several transcription factors directing the heat-inducible transcription of target genes, for instance, heat shock factor (HSF) [16,17,18], multiprotein bridging factor 1 (MBF1) [19,20], and DEHYDRATION-RESPONSIVE ELEMENT BINDING PROTEIN 2A (DREB2A) [21,22]. Other studies have focused on the trigger of the heat-inducible expression of *HSP* genes, which is related to the elucidation of how photosynthetic organisms sense heat stress [23,24,25]. This approach revealed the induction of *HSP* gene expression by heat-induced membrane fluidization in prokaryotes and eukaryotes [23,24,25,26,27,28,29,30,31,32], thus raising the possibility that organisms sense membrane fluidization as a signal of heat stress [33,34,35]. Ca^2+^ channels on cellular membranes have been proposed to act as heat sensors in terrestrial plants, such as the moss *Physcomitrium patens* and Arabidopsis (*Arabidopsis thaliana*) [36,37,38]. By contrast, *HSP* expression does not depend on membrane fluidization in the unicellular green alga *Chlamydomonas reinhardtii* [39]; thus, it is unclear whether the heat-inducible expression of *HSP* genes completely depends on membrane fluidization in other eukaryotes.

*HSP70* genes have been identified in marine multicellular algae called seaweeds, including the green alga *Ulva porfiera* [40]; the brown algae *Fucus* spp. [41] and *Fucus distichus* [42]; and the Bangiales order of red algae that includes *Porphyra umbilicalis* [43,44], *Neopyropia yezoensis* [45,46], and *Neoporphyra haitanensis* [47]. Most of these *HSP* genes were found to be heat inducible. A distinctive feature of CP-localized HSP70s in the Bangiales is that they are encoded by *DnaK* genes in the chloroplast genome [43,48,49]; however, the expression profiles of these genes are unknown. A comparison of the mechanisms regulating the transcription of the *HSP70* genes in nuclear and chloroplast genomes is necessary to understand the unique heat stress response in the Bangiales.

We recently analyzed the importance of the membrane physical state for the induction of heat-inducible gene expression in *N. yezoensis* [50], revealing that membrane fluidization by heat stress induced the expression of two cytoplasmic *HSP70*s and *MBF1* but not the newly identified *HIGH TEMPERATURE RESPONSE 2* (*HTR2*) or *HTR2 LIKE* (*HTR2L*). This observation indicates that *N. yezoensis* possesses membrane fluidization-dependent and -independent signal transduction pathways in its heat stress response, suggesting the presence of an additional unknown trigger for heat-inducible gene expression. These findings prompted us to question whether the nuclear genes for ER- and MT-localized HSP70 proteins and the chloroplast gene for CP-localized HSP70 are heat inducible and, if so, whether the trigger(s) for the heat-inducible expression of these nuclear and plastid genes is common or distinct in *N. yezoensis*.

In the present study, we identified the genes for ER- and MT-localized HSP70s in *N. yezoensis* and confirmed the heat inducibility of these nuclear genes and the previously identified cytoplasmic and plastid *HSP70* genes. In addition, we determined the contribution of membrane fluidization in the heat-inducible expression of all *HSP* genes. Our findings reveal the trigger of the heat-induced expression of the *HSP70* genes in *N. yezoensis* and provide novel and unique insights into the coordinated expression of the nuclear and plastid genomes, as well as the diversification of the regulatory mechanisms of the heat stress response in photosynthetic organisms.

## 2. Results

### 2.1. Identification of the Genes for ER and MT HSP70 in N. yezoensis

Using a functional annotation of our previous transcriptome data [51], we identified five contigs and unigenes (CL1648.Contig2, CL2684, Unigene14751, Unigene14752, and Unigene2869) representing candidate *HSP70* genes. To classify these genes, we performed a phylogenetic analysis using the amino acid sequences encoded by these genes and HSP70s from unicellular red algae and the Bangiales. We included HSP110 members, which are related to cytosolic HSP70s but exhibit unique structural characteristics in the ATPase- and C-terminal domains [52,53,54], as the outgroup. We also included the previously identified CP HSP70, designated NycpDnaK, encoded by the chloroplast genome of *N. yezoensis* (GenBank accession number: DQ497595).

The translational products of all candidate genes contained three HSP70 family signatures [55] in their N-terminal ATPase domains (Appendix A); thus, we concluded that these are bona fide *HSP70* genes in *N. yezoensis*. As shown in Figure 1, these proteins clustered into two major clades, each consisting of two subclades. The previously identified genes *NyHSP70-1* and *NyHSP70-2* [45,50], which correspond to Unigene14751 and CL1648.Contig2, respectively, grouped in the same subclade. Members of this subclade contain the cytoplasmic HSP70-specific motif EDID or SEVD (Appendix A), taking on the typical sequence of EEVD in the HSP70s of other eukaryotic organisms [54,55]. This subclade is therefore a distinct class of cytoplasmic HSP70s. The other subclade in the same major clade contained the proteins encoded by Unigene2869 and CL2684. These proteins contain a putative ER-retention tetrapeptide, K/HEEL (typically K/HDEL) [54,55]; therefore, we conclude that this subclade is for ER-localized HSP70 (also named binding protein [BiP]), and we designated them *NyBiP1* and *NyBiP2*, respectively (Figure 1). We detected a 43-amino acid signal peptide (MAATPGSPARRPRPAAAKRRSVWTAVLVAAASIATATVAVSAA) in NyBiP1, while BiP2 did not appear to harbor a signal peptide.

Chloroplast genome-encoded NycpDnaK and proteins included in the same subclade possessed the motif TVIDTDFSEAK (Appendix A), which is related to the DVIDADFTDSK motif conserved in chloroplast HSP70s [54]. This subclade therefore contains the CP-localized HSP70s. The proteins in the rest of the subclade had characteristic motifs, such as GDAWVS and YSPSQV (Appendix A), which are conserved signature sequences for MT-localized HSP70s [54]. We thus designated Unigene14751 in this subclade *NymtDnaK*. Its encoded protein had a 33-amino acid signal peptide (MASAVSARAALARVGACRPGLLGLAASGGGGLA).

These findings indicate that the Bangiales, including *N. yezoensis* and unicellular red algae, possess all four classes of HSP70s, as well as HSP110, found in terrestrial plants [14,54,56,57]. The ER and MT HSP70s are newly identified; we deposited the sequence information of NyBiP1, NyBiP2, and NymtDnaK to DDBJ/EMBL/GenBank (accession numbers LC764584, LC764585 and LC764586, respectively). The NyHSP110 sequence was also deposited (LC764587).

### 2.2. Heat-Inducible Expression of the ER, CP, and MT HSP70 Genes

Our previous study demonstrated that when algal samples were cultured under a 10 h light and 14 h dark photoperiod at 15 °C, the expression of NyHSP70-1 and NyHSP70-2 was induced transiently by a treatment of 25 °C, peaking 2 h after the onset of temperature increase and again after 1 h of darkness (8 h after the beginning of the higher temperature treatment) [50]. We therefore analyzed the expression profiles of genes for the ER-, CP-, and MT-localized HSP70s in the three life stages—the gametophyte, sporophyte, and conchosporophyte [51]—under these conditions using the gene-specific primer sets indicated in Appendix A.

As shown in Figure 2, NyBiP1 and NyBiP2 expression was transiently induced twice as much as observed for cytoplasmic HSP70 genes when algae were grown at 25 °C but not at 5 °C in all three life stages, indicating the transient heat-inducible and dark-stimulated expression of these genes. However, we observed differences in the expression profiles of the genes: NyBiP1 was mainly induced by 25 °C with a peak at 2 h into the higher temperature in the sporophyte and conchosporophyte, whereas NyBiP2 was mainly induced by darkness under heat stress conditions in all three life stages (8 h after onset of higher temperature and 1 h of darkness). In addition, the expression of two organellar HSP70 genes, NycpDnaK and NymtDnaK, was induced at 25 °C in the light and darkness at 25 °C, with expression peaks at 2 and 8 h into the temperature increase, although NycpDnaK was mainly induced by darkness and its expression level in the conchosporophyte remained low (Figure 3). Furthermore, when the algal samples were cultured under continuous light to explore the factor required for the second expression peak, we noticed that the dark-stimulated expression of the genes for ER- and organelle-localized HSP70 proteins is commonly inhibited by light in all three life stages (Appendix A). This result indicates that darkness, and not the duration since onset of heat treatment, is the factor required for the expression of the organellar HSP70-encoding genes under heat stress conditions, as was also shown previously for the genes encoding two cytoplasmic HSP70s [50].

### 2.3. Contribution of Membrane Fluidization to the Heat-Inducible and Dark-Stimulated Expression of HSP70 Genes

Our previous findings demonstrated the presence of two different heat stress signal transduction cascades, the membrane fluidization-dependent and -independent pathways, and the involvement of the former in NyHSP70-1, NyHSP70-2, and NyMBF1 expression [50]. To determine which pathway regulates the heat-inducible expression of the genes for ER- and organelle-localized HSP70 proteins, we examined the effects of physiological membrane changes on the expression of these genes using a membrane fluidizer (benzyl alcohol; BA) and a membrane rigidizer (dimethyl sulfoxide; DMSO). The treatment with 2.5 mM BA, but not 4% (*v/v*) DMSO, induced the expression of all four genes at 15 °C with similar expression kinetics, with an expression peak observed 15 min after the onset of pharmacological treatment (Figure 4 and Figure 5). The heat-induced gene expression for ER- and organelle-localized HSP70 is thus regulated by the membrane fluidization-dependent pathway.

We asked whether the second peak seen under darkness and heat conditions requires membrane fluidization. Accordingly, we examined the effects of 2.5 mM BA on the expression of the genes for ER- and organelle-localized HSP70 under continuous light in the three life stages at 15 °C. We determined that BA induces the expression of all genes at 1 and 5 h into treatment in the three life stages (Figure 6). We also subjected the three life stages of the alga to heat stress, either with or without 4% DMSO, under dark conditions. This DMSO treatment decreased the dark-stimulated expression of all genes (Figure 7). Thus, darkness under heat stress conditions further fluidizes the membranes whose state of the fluidity has already adapted to 25 °C under the light, resulting in the promotion of the membrane fluidization-dependent induction of the HSP70 genes. These findings indicate that the dark-stimulated expression of the genes encoding ER and organelle-localized HSP70 is regulated by a pathway activated by membrane fluidization in *N. yezoensis*.

### 2.4. Effects of Temperature Changes on the Expression of Plastid Genes for the Two-Component System

Red algal plastid genomes contain genes encoding one histidine kinase (Ycf26) and four response regulators (Ycf27, Ycf28, Ycf29, and Ycf30) [58]. Of these, Ycf26 is proposed to regulate the transcription of the chloroplast gene NycpDnaK, as its homolog in the photosynthetic cyanobacterium Synechocystis sp. PCC 68803, Hik33, contributes to inducing gene expression in an environmental stress-dependent manner [59,60].

To explore this possibility, we examined the effects of temperature changes on the expression of genes encoding Ycf26 and Ycf27, a response regulator proposed to work together with Ycf26 as a two-component system. We detected no response for Ycf26 or Ycf27 expression to either 5 °C (cold stress) or 25 °C (heat stress) in the three life stages (Appendix A). Thus, the transcriptional induction of these genes might not be involved in the heat-inducible expression of NycpDnaK.

## 3. Discussion

HSP70 is conserved in a wide variety of organisms [8,9,10], and the regulatory mechanisms underlying the heat-inducible expression of the encoding genes have been extensively studied [13,14,15,16,17,18]. Despite these insights, the analysis of heat-inducible transcription of the *HSP70* genes in the Bangiales has been limited to genes encoding cytoplasmic HSP70 in *N. yezoensis* [45,50]. Here, we identified new members of the *HSP70* family and analyzed their expression profiles under heat stress conditions in the marine red alga *N. yezoensis*. A phylogenetic analysis indicated that the family members were divided into four classes, cytoplasmic HSP70, ER-localized HSP70 (BiP), CP-localized HSP70, and MT-localized HSP70, as observed in other eukaryotes [12,13]. In addition, genes for ER- and organelle-localized HSP70s display heat-induced and dark-stimulated expression in the three studied life stages. Genes for cytoplasmic HSP70 also displayed these characteristics [50], leading us to conclude that the expression of all *HSP70* genes in *N. yezoensis* is heat inducible in the three life stages studied, as previously reported for *NyMBF1* [50]. Therefore, our study demonstrates for the first time that ER- and organelle-localized HSP70 proteins are encoded by heat-inducible genes whose expression is regulated by the membrane fluidization-dependent signal transduction pathway in the red alga *N. yezoensis*. Together with our previous data regarding cytoplasmic HSP70 [50], we conclude that genes of all four *HSP70* classes are commonly and coordinately induced by heat stress in a membrane fluidization-dependent manner.

Recently, a genome-wide survey of the *HSP70* genes in *N. yezoensis* led to the identification of 15 genes [46], although these included *HSP110* genes, the number may not be accurate. In their study, four classes of *HSP70*s were proposed, as in our present study; however, their phylogenetic tree was different from the one presented in Figure 1. Our tree clearly demonstrated the separation of cytoplasm- and ER-localized HSP70s from organellar HSP70s, revealing differences in the origins of the organellar HSP70s and the other HSP70s [14,15]. By contrast, the phylogenetic tree produced by Yu et al. [46] included ER-localized HSP70s in the clade containing organellar HSP70s, and one of the CP-localized HSP70s was located in the cytoplasmic HSP70s clade. It is not clear why such a difference occurred; thus, we must identify other *HSP70* genes in *N. yezoensis* and revise the phylogenetic trees to validate their true relationships and origins. Moreover, Yu et al. [46] reported the presence of two nuclear genes encoding chloroplast-localized HSP70 in *N. yezoensis*. They used established analysis tools to predict the subcellular localization of HSP70s; however, it remains necessary to identify the conserved amino acid sequence motifs and putative chloroplast signal peptides to resolve the uncertain classification and characterization of the *N. yezoensis* HSP70s.

*NycpDnaK* is characteristically a chloroplast gene in red algae [43,48,49], while this chloroplast-localized HSP70 is usually a nuclear gene in other eukaryotes [61,62,63]. The red algae genes encoding the chaperonin GroES and two-component histidine kinase and response regulators have also been retained in the chloroplast genome, while their homologs in terrestrial plants are in the nuclear genome. The red algal chloroplast genome thus appears to be in an evolutionarily intermediate state, with an incomplete transfer of genes from the ancient symbiotic chloroplast genome to the nuclear genome. The maintenance of these genes in the organelle may provide functional benefits in red algae, however, enabling them to survive in the ever-changing conditions of the intertidal region. In the presence of chloroplast-targeted DnaJ proteins, for example, the heat-inducible expression of *NycpDnaK* might be faster from the chloroplast than the nuclear genome, rapidly producing HSP70s (and the HSP70/HSP40 chaperone complex) to protect and fine-tune various organellar proteins. This hypothesis might be supported by the presence of *GroEL* and *GroES* as an operon in the chloroplast genomes of the Bangiales, facilitating the rapid supply of the HSP60/HSP10 chaperonin complex in this organelle, although it is uncertain whether the expression of the *GroEL*-*GroES* operon is heat inducible. To understand why *NycpDnaK* and *GroES* are located in the chloroplast genome of the Bangiales, the regulatory mechanisms of their transcription should be analyzed under various environmental stress conditions.

Although membrane fluidization is a common trigger of the heat-inducible expression of *N. yezoensis HSP70* genes, the regulatory mechanisms of heat-inducible transcription are proposed to be different between nuclear and chloroplast genes. The involvement of HSF, a transcription factor responsible for controlling the heat-inducible expression of HSP genes, has been confirmed in many eukaryotes [64,65]. Red algae also have HSF [66,67], but the identification and expression analysis of the HSF gene(s) have yet to be conducted in *N. yezoensis*. In addition, the expression of chloroplast genes is regulated by nucleus-encoded transcription factors called sigma factors, which contribute to the differential expression of chloroplast genes under various environmental stress conditions. In this study, we discovered that the expression of genes encoding histidine kinase and the response regulator were not induced by heat stress (Appendix A); however, it is highly possible that the heat-dependent activation of Ycf26 by histidine phosphorylation and the phosphorylation of Ycf27 by activated Ycf26 are involved in NycpDnaK expression. The identification of transcription factors and the elucidation of the functional mode of the chloroplast two-component system are therefore essential for understanding the regulatory mechanisms underlying the coordinated heat-inducible gene expression for nuclear and chloroplast HSP70 genes in *N. yezoensis*.

The involvement of membrane fluidization in heat-inducible gene expression suggests the presence of a heat sensor that perceives heat-dependent changes in the membrane physical state and acts as an initiator to activate heat signal transduction pathways that govern the expression of all *HSP* genes in *N. yezoensis*. It has been proposed that membrane-localized calcium ion (Ca^2+^) channels are heat sensors in terrestrial plants [36,37,38,68]. Indeed, many lines of evidence indicate the contribution of Ca^2+^ influxes in the early phase of the heat stress response in terrestrial plants [36,37,38,68,69,70]; in moss, the activation of plasma membrane-embedded cyclic nucleotide-gated Ca^2+^ channels depends on membrane fluidization [36,37,38,68]. Thus, it is plausible that Ca^2+^ channels might be heat sensors that become activated by the heat-induced membrane fluidization in *N. yezoensis*; however, little is currently known about the heat sensor and heat stress signal transduction pathways. Understanding how membrane fluidization activates the signal transduction pathways that target the expression of nuclear and chloroplast *HSP70* genes will therefore require the elucidation of whether Ca^2+^ influx is involved in either or both signal transduction pathways. These experiments could enable us to examine the relationship between Ca^2+^ influx and the expression and activation of transcription factors such as HSFs and sigma factors in red algae. The resulting insights may permit the identification of the heat sensor and the elucidation of the regulatory mechanisms involved in the recognition and transmission of the membrane fluidization signal to different signaling pathways to regulate the heat-inducible expression of the *N. yezoensis HSP70* genes in a coordinated manner.

Together with our previous results [50], our findings demonstrate the presence of two different cascades for regulating heat-inducible gene expression in *N. yezoensis*. One is the membrane fluidization-dependent pathway that regulates multiple HSP70 and *NyMBF1* genes, which are part of different sub-pathways for the regulation of heat-inducible nuclear and chloroplast genes as a unique system in the Bangiales. The other is the membrane fluidization-independent pathway for the expression of genes encoding the uncharacterized proteins NyHTR2 and NyHTR2L [50]; the trigger for the activation of this pathway is unknown. Signaling factors such as Ca^2*+*^ and the transcription factors involved in each pathway must be elucidated to identify the heat sensors and illuminate the entire regulatory system of the heat stress response mediated by multiple heat signal transduction pathways with their own heat-inducible target genes in *N. yezoensis*.

## 4. Materials and Methods

### 4.1. Algal Materials and Culture Conditions

Artificial sterilized seawater (SEALIFE, Marinetech, Tokyo, Japan) enriched with ESS2 [71] was used for the maintenance and examination of the three life stages of *N. yezoensis* (strain U-51)—gametophytes, sporophytes, and conchosporophytes [51]—under 60–70 μmol m^–2^ s^–1^ of light with a short-day photoperiod (10 h light/14 h dark) at 15 °C. Air was filtered through a 0.22 μm filter (Whatman, Maidstone, UK). The culture medium was changed weekly.

Heat stress and pharmacological treatments were performed as described by Khoa and Mikami [50]. In brief, the three life stages were incubated at 5 °C, 15 °C, or 25 °C for 0.5, 1, 2, 4, 6, 8, and 12 h. The experiments started at 12:00, 3 h after the start of light irradiation (9:00). In the short-day photoperiod with 10 h light, the 8 and 12 h time points comprised 1 and 5 h of darkness, respectively. Continuous 24 h lighting was also employed as a control condition. Treatments with 2.5 mM of benzyl alcohol (BA) and 4% dimethyl sulfoxide (DMSO) were performed by incubating algae at 15 °C for 5, 15, or 30 min. Algal cells were also treated with BA at 25 °C in the light or with DMSO under dark conditions for 8 or 12 h. After these treatments, algae were frozen in liquid nitrogen and stored at −80 °C prior to their use for gene expression analysis.

### 4.2. Identification and Phylogenetic Analysis of HSP70 Genes

Contigs and unigenes annotated as *HSP70* genes in our *N. yezoensis* transcriptome data [51] were examined with open reading frame (ORF) finder (https://www.ncbi.nlm.nih.gov/orffinder/ accessed on 10 March 2023) to identify full-length coding sequences. A BLAST search (https://blast.ncbi.nlm.nih.gov/Blast.cgi) using the predicted amino acid sequences for their translated products was performed to validate their HSP70 identity. The presence of a signal peptide was examined using the SOSUIsignal program (https://harrier.nagahama-i-bio.ac.jp/sosui/sosuisignal/sosuisignal_submit.html accessed on 15 April 2023). The amino acid sequences of all obtained HSP70 homologs were combined with previously known red algal HSP70s to reconstruct a neighbor-joining phylogenetic tree in MEGA 7 (https://www.megasoftware.net), using ClustalW to align the sequences.

### 4.3. Total RNA Extraction, DNA Removal, cDNA Synthesis, and Quantitative Gene Expression Analysis

All procedures were performed as described by Khoa and Mikami [50], except that primers for the *HSP70* genes were used (Appendix A) for reverse transcription quantitative PCR (RT-qPCR). Total RNAs from the three stages were separately extracted using a FavorPrep Plant Total RNA Mini Kit (FAVORGEN, Ping Tung, Taiwan) and subjected to treatment with DNase I (TURBO DNA-free TM kit; Thermo Fisher Scientific, Waltham, MA, USA) to remove any genomic DNA contamination. Total RNA samples (300 ng) with A260/A280 ratios ranging from 1.9 to 2.1 were used to synthesize first-strand complementary DNA (cDNA) using a PrimeScript 1st strand cDNA Synthesis Kit (Takara Bio, Kusatsu, Japan). The thermal cycling parameters consisted of an initial denaturation step at 98 °C for 30 s, followed by 30 cycles of 98 °C for 10 s, 60 °C for 30 s, and 72 °C for 20 s and a final extension step at 72 °C for 5 min.

The primers for the *HSP70* genes were designed using Primer Premier 5 (http://www.premierbiosoft.com accessed on 11 December 2021). The sizes of the amplified products and the specificity of the primers were confirmed using PCR reactions with a mixture of three cDNA samples and each primer set, using a Phusion high-fidelity DNA polymerase and GC buffer (Biolabs, Boston, MA, USA), according to the manufacturer’s instructions. The PCR products were checked by agarose gel electrophoresis. Primer sets generating amplified DNA bands with the expected sizes were employed for the qPCR, which was carried out in a total volume of 20 μL, containing 10 μL of 2× SYBR Premix Ex Taq GC, 0.4 μL of ROX Reference Dye, 2 μL of cDNA template, and 0.4 μL (10 μM) of each primer, using a SYBR Premix Ex Taq GC kit (Takara Bio). The thermal cycling parameters for the AriaMX (3000P) real-time PCR system (Agilent Technologies, Santa Clara, Hercules, CA, USA) consisted of 95 °C for 3 min and 40 cycles of 95 °C for 5 s and 60 °C for 20 s. A dissociation curve was generated to check for the specificity of amplification using the following conditions: 95 °C for 1 min, 55 °C for 30 s, and 95 °C for 30 s. The translation initiation factor 4A (*eIF4A*) gene was employed as a reference gene [50].

### 4.4. Statistical Analysis

Values are presented as means ± standard deviation (SD) from triplicate experiments. A one-way ANOVA followed by a Tukey–Kramer test was used for multiple comparisons, and significant differences were determined using a cutoff value of *p* < 0.05.

## Figures and Tables

**Figure 1 plants-12-02070-f001:**
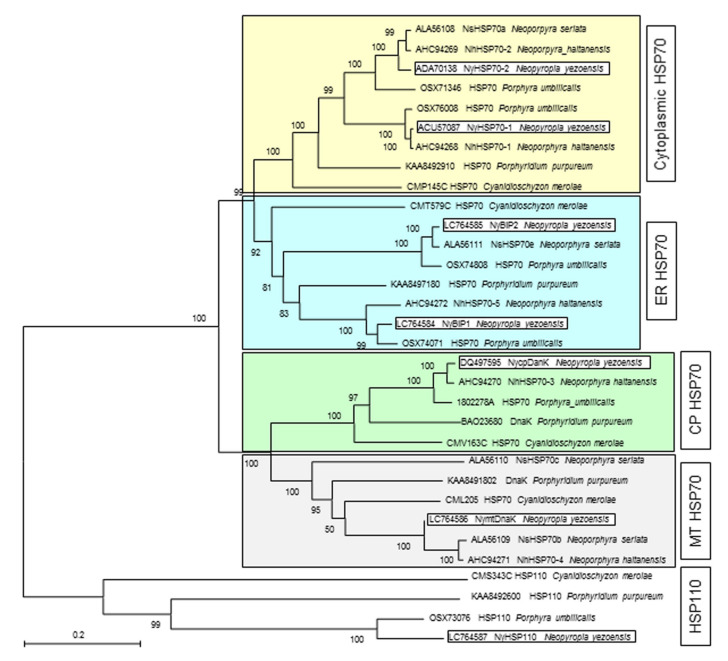
Phylogenetic analysis of HSP70s from the Bangiales and unicellular red algae. Each DDBJ/EMBL/GenBank accession number is indicated before the species names in the tree. Red algal HSP110 proteins were used for the outgroup. The cytoplasmic, ER, CP, and MT HSP70 clades and the HSP110 outgroup are indicated by colored boxes. The HSP70 homologs from *N. yezoensis* are boxed. The percentage support for each branch from 1000 bootstrap replicates is indicated at the nodes of the tree. The scale bar indicates 0.2 substitutions per site.

**Figure 2 plants-12-02070-f002:**
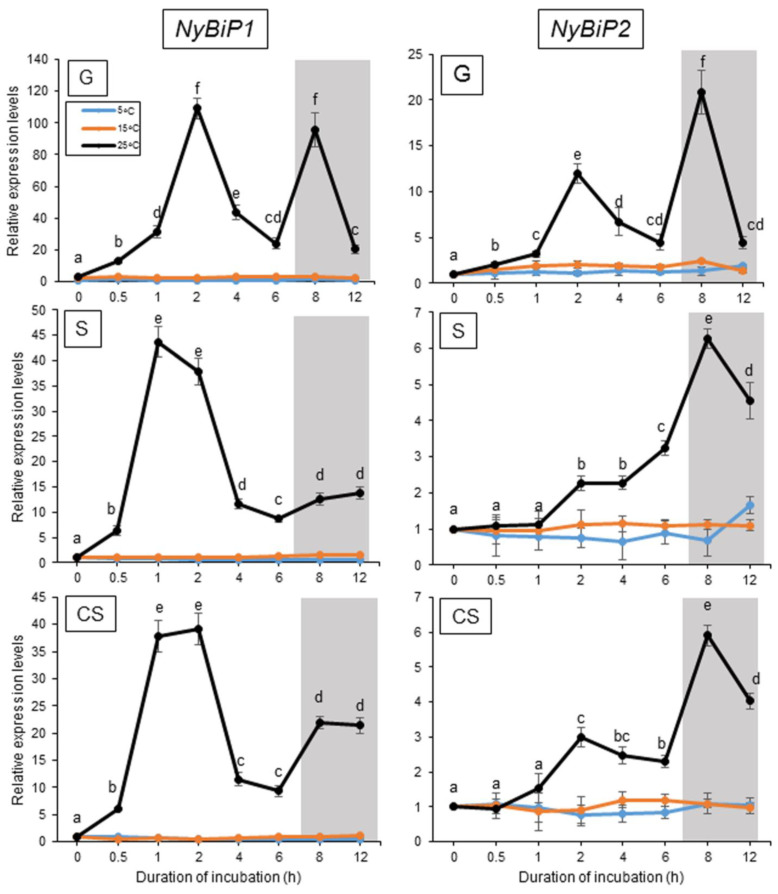
Effects of temperature changes on the expression of *NyBiP1* and *NyBiP2* in three life stages of *Neopyropia yezoensis*. Values on the *y*-axis represent the fold-change of the expression of each gene relative to that at 0 h. Shading indicates dark period. Significant differences in the expression level in the three life stages, indicated by different letters, were defined from triplicate independent replicate data using a one-way ANOVA with a Tukey’s test (*p* < 0.05). G, gametophyte; S, sporophyte; CS, conchosporophyte.

**Figure 3 plants-12-02070-f003:**
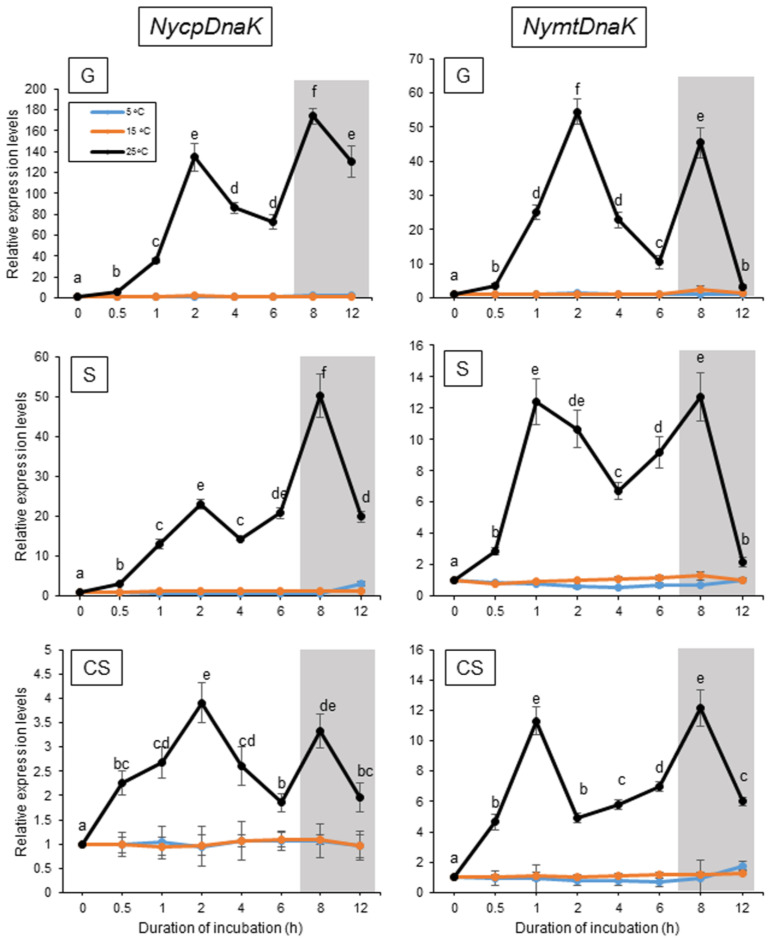
Effects of temperature changes on the expression of *NycpDnaK* and *NymtDnaK* in three life stages of *Neopyropia yezoensis*. Values on the *y*-axis represent the fold-change of the expression of each gene relative to that at 0 h. Shading indicates dark period. Significant differences in the expression level in the three life stages, indicated by different letters, were defined from triplicate independent replicate data using a one-way ANOVA with a Tukey’s test (*p* < 0.05). G, gametophyte; S, sporophyte; CS, conchosporophyte.

**Figure 4 plants-12-02070-f004:**
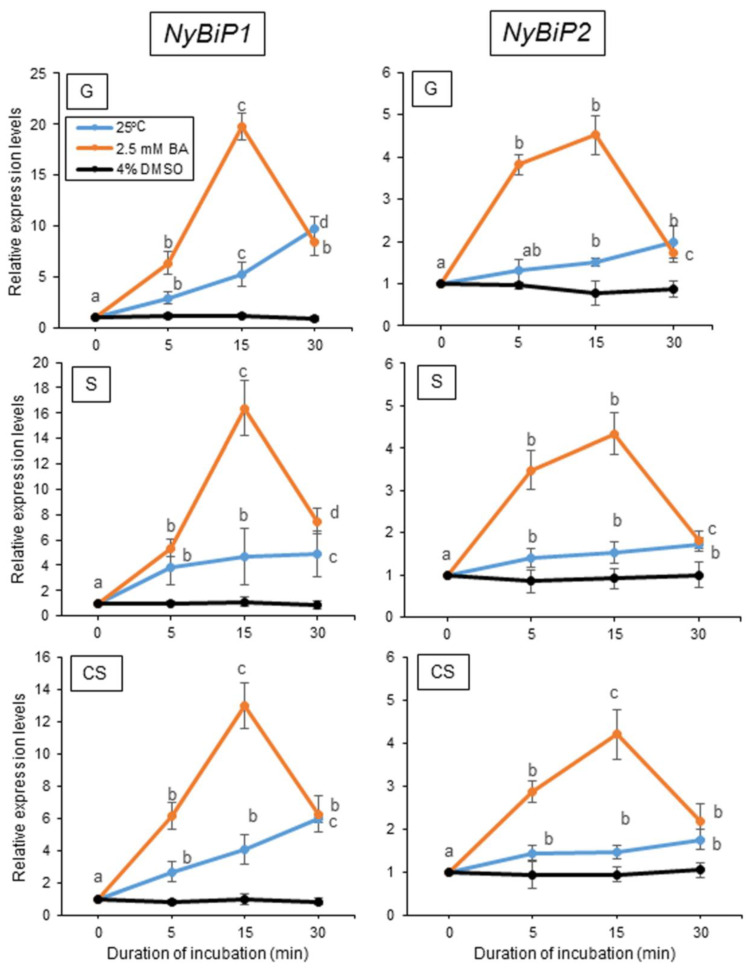
Relationship between the changes in the physical state of the membrane and the expression of *NyBiP1* and *NyBiP2* in three life stages of *Neopyropia yezoensis*. Membrane fluidization or rigidification was induced by pharmacological treatment with 2.5 mM benzyl alcohol (BA) or 4% dimethyl sulfoxide (DMSO), respectively, at 15 °C for 5, 15, and 30 min, after which the expression of the genes of interest was compared to their expression at 25 °C. Values on the *y*-axis represent the fold-change in the expression of each gene relative to that at 0 h. Significant differences in the expression level in the three life stages, indicated by different letters, were defined from triplicate independent replicate data using a one-way ANOVA with a Tukey’s test (*p* < 0.05). G, gametophyte; S, sporophyte; CS, conchosporophyte.

**Figure 5 plants-12-02070-f005:**
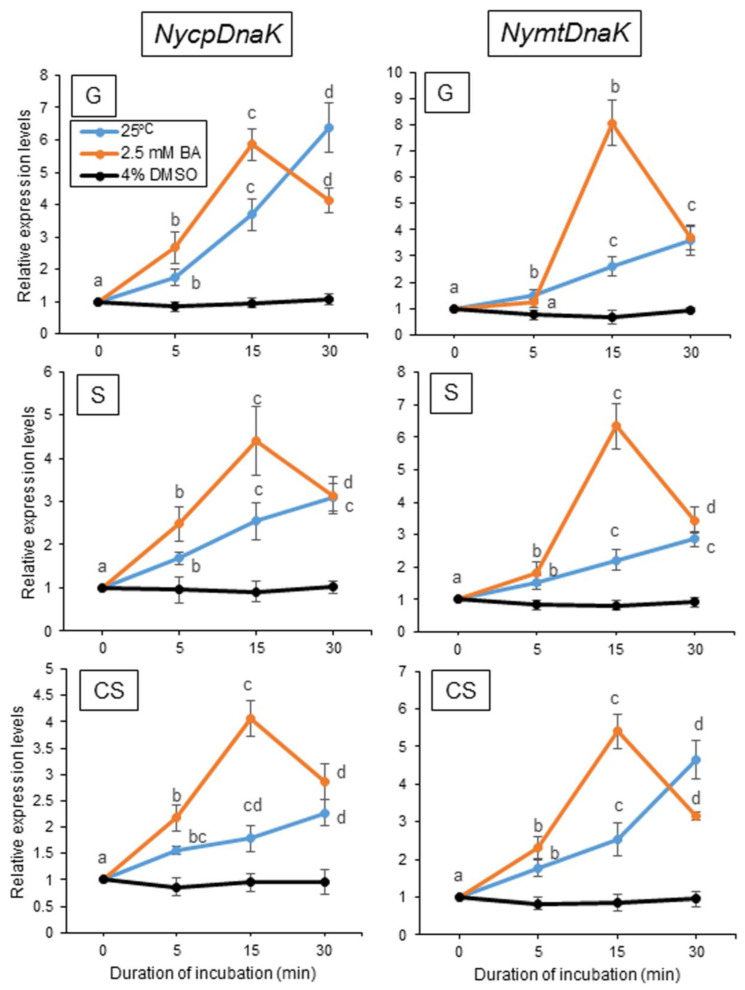
Relationship between the changes in the physical state of the membrane and the expression of *NycoDnaK* and *NymtDnaK* in three life stages of *Neopyropia yezoensis*. Membrane fluidization or rigidification was induced by pharmacological treatment with 2.5 mM benzyl alcohol (BA) or 4% dimethyl sulfoxide (DMSO), respectively, at 15 °C for 5, 15, and 30 min, after which the expression of these genes was compared to their expression at 25 °C. Values on the *y*-axis represent the fold-change in the expression of each gene relative to that at 0 h. Significant differences in the expression level in the three life stages, indicated by different letters, were defined from triplicate independent replicate data using a one-way ANOVA with a Tukey’s test (*p* < 0.05). G, gametophyte; S, sporophyte; CS, conchosporophyte.

**Figure 6 plants-12-02070-f006:**
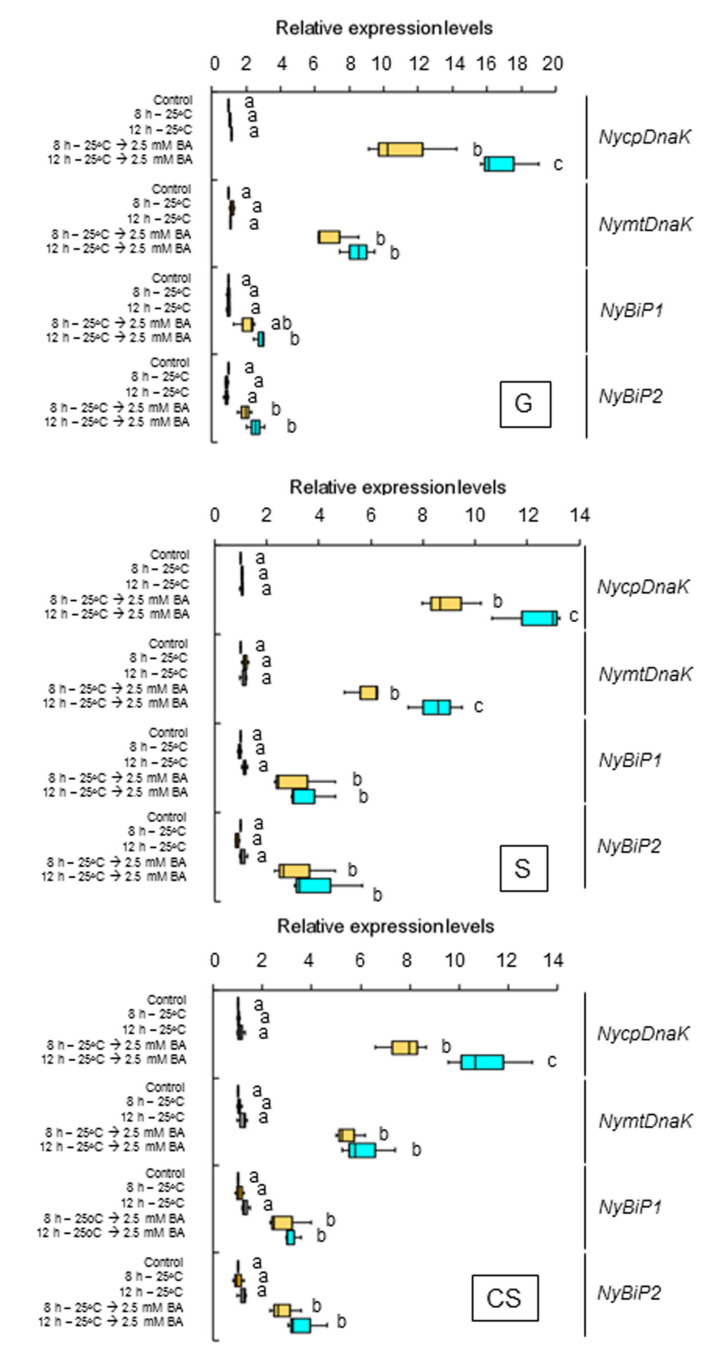
Requirement for membrane fluidization in the dark-induced second gene expression peaks in the three life stages of *Neopyropia yezoensis*. The expression of genes for ER and organellar *HSP70* was examined at 8 and 12 h into heat stress (25 °C) exposure under continuous light with or without a treatment of 2.5 mM BA. Values on the *y*-axis represent the relative fold-change in the expression of each gene. Significant differences in the expression level in the three life stages, indicated by different letters, were defined from triplicate independent replicate data using a one-way ANOVA with a Tukey’s test (*p* < 0.05). G, gametophyte; S, sporophyte; CS, conchosporophyte.

**Figure 7 plants-12-02070-f007:**
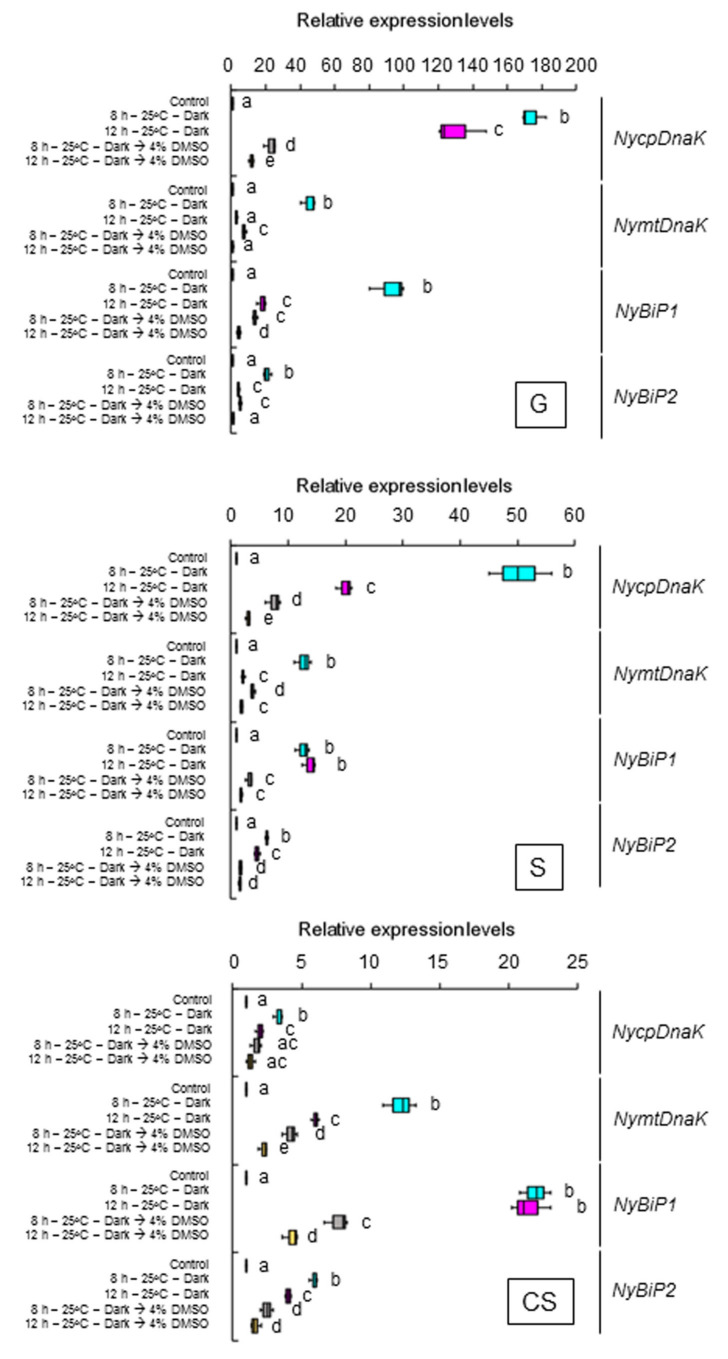
Effects of membrane rigidification on the dark-induced second gene expression peaks in the three life stages of *Neopyropia yezoensis*. The expression of genes encoding ER and organellar HSP70 was examined at 8 and 12 h into heat stress (25 °C) exposure under continuous light with or without a treatment of 4% DMSO. Values on the *y*-axis represent the relative fold-change in the expression of each gene. Significant differences in the expression level in the three life stages, indicated by different letters, were defined from triplicate independent replicate data using a one-way ANOVA with a Tukey’s test (*p* < 0.05). G, gametophyte; S, sporophyte; CS, conchosporophyte.

## Data Availability

Data are contained within the article.

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
