# Peer review of "Membrane Fluidization Governs the Coordinated Heat-Inducible Expression of Nucleus- and Plastid Genome-Encoded Heat Shock Protein 70 Genes in the Marine Red Alga Neopyropia yezoensis"

_plants, 2023, doi:10.3390/plants12112070_

Round 1

Reviewer 1 Report

The scientists’ interest in heat shock protein 70 has not waned for many years. Employment of marine red alga allows clarifying the regulation and functions of these important proteins. Authors demonstrate heat-induced and dark-stimulated expression of the ER- and organelle-localized HSP70s, and show that the membrane fluidization is a trigger for the coordinated induction of HSP70 genes not just in nuclear, but also in plastid genome of N. yezoensis.

The manuscript is written well, using clear precise language. The structure of this article and the presentation of the data highly resemble the previously published work “Khoa, H.V.; Mikami, K. Membrane-Fluidization-Dependent and -Independent Pathways Are Involved in Heat-Stress-Inducible Gene Expression in the Marine Red Alga Neopyropia yezoensisCells 202211, 1486”

It is highly desirable to present data confirming the specificity of effects of benzyl alcohol and DMSO treatment by including the expression data for the genes not affected (for BA) and affected (for DMSO) by the treatment.

Minor comments

The asterisks, colons, and periods are not shown under the HSP70 signature sequences of Figure S1.

Line 123 Please, correct “Figyre 1”

Please, provide the information about the biological significance of the dark induction of the HSP70s expression.

Please, indicate that BA is benzyl alcohol earlier in the text, when is mentioned for the first time.

Line 194. Should it be “Figure 2”?

Supplemental Fig. 2 – the color indication is misleading. Please, indicate what does the background color mean.

Please, correct  “Fig.S3.Effects of tMemperature changes”

Why do the authors think that “darkness under heat stress conditions further fluidizes the membrane”?

Please, include short description of OmpR-type response.

Technical errors have to be corrected.

Author Response

The asterisks, colons, and periods are not shown under the HSP70 signature sequences of Figure S1.

Response: Please recheck the submitted figure where the asterisks, colons, and periods are certainly indicated.

Line 123 Please, correct “Figyre 1”

Response: Thank you very much for your suggestion. We corrected it to “Figure 1” at line 123 in the revised manuscript.

Please, provide the information about the biological significance of the dark induction of the HSP70s expression.

Response: Please see my reply for your last comment.

Please, indicate that BA is benzyl alcohol earlier in the text, when is mentioned for the first time.

Response: Thank you very much for your suggestion. We added “benzyl alcohol” at line 220 in the revised manuscript. Accordingly, “2.5 mM BA or 4% DMSO” in legends to Figures 4 and 5 was revised to “2.5 mM benzyl alcohol (BA) or 4% dimethyl sulfoxide (DMSO)” (lines 178 and 186 in the revised manuscript).

Line 194. Should it be “Figure 2”?

Response: Since this part have to represent the data for NycpDnaK and NymtDnaK, Figure 3 is correct, that is different from Figure 2 showing the data for NyBiP1 and NyBiP2.

Supplemental Fig. 2 – the color indication is misleading. Please, indicate what does the background color mean.

Response: As shown in the box under the graphs, each color represented duration of the incubation under the light or dark. This figure indicated the dark-requirement of the second peak at 8 and/or 12 h after starting the heat treatment. Indeed, continual light for 8 and 12 h completely inhibited the induction of gene expression. Therefore, we are sorry we cannot get your point.

Please, correct  “Fig.S3.Effects of tMemperature changes”

Response: Thank you very much for your suggestion. We provided the new version of Figure S3.

Why do the authors think that “darkness under heat stress conditions further fluidizes the membrane”?

Response: To clarify our proposal, we added the sentence “whose state of the fluidity has already adapted to 25℃ under the light” at lines 233-234.

Please, include short description of OmpR-type response.

Response: We deleted the word “OmpR-type”.

Reviewer 2 Report

The manuscript submitted by Mikami and Khoa describes the identification of a new member of the HSP70 family in the marine red alga N. yezoensis and the analysis of its expression under heat stress. The authors found that organelle-localized HSP70 expression was induced by heat stress treatment and dark stimulations at three different life stages. In addition, they identified two types of cascades, membrane fluidization-dependent and -independent, as pathways regulating heat-inducible gene expression. The research is well-designed and topics described are interesting. However, I would like to point out the following issues that are in need of attention and correction.

Figure 2. For timescale data, I would think a line chart would be appropriate, not a box plot. It would be easier to understand if the x-axis were to represent time points including light and dark periods, and different temperature treatments were represented by different colored line charts. Figures 3, 4, and 5 are also.

In this submitted manuscript, ORF and peptide sequences have been analyzed. I am wondering whether any conserved cis-regulatory elements can be found in the promoter region of the HSP70 genes, i.e. membrane fluidization-dependent and -independent regulatory mechanisms. It should be discussed.

The writing style and English of the manuscript has to be checked by a professional editor.

Author Response

Figure 2. For timescale data, I would think a line chart would be appropriate, not a box plot. It would be easier to understand if the x-axis were to represent time points including light and dark periods, and different temperature treatments were represented by different colored line charts. Figures 3, 4, and 5 are also.

Response: We changed the style of Figures 2-5 according to your comments. Dark period was also indicated in Figures 2 and 3, by which legends of these figures were added a sentence “Shading indicates dark period”.

In this submitted manuscript, ORF and peptide sequences have been analyzed. I am wondering whether any conserved cis-regulatory elements can be found in the promoter region of the HSP70 genes, i.e. membrane fluidization-dependent and -independent regulatory mechanisms. It should be discussed.

Response: Yes, our experiments were based on the sequence information of ORFs. We also have interests on the regulatory mechanisms of membrane fluidization-dependent and –independent gene expression. Thus, in near future, we hope to examine promoter regions of HSP family genes in Neopyropia yezoensis. At present, there is no information about promoters and thus we cannot discuss anything about regulatory elements and their cognate transcription factors. In fact, cis-regulatory elements involved in membrane-fluidization-dependent transcription has not identified in other organisms to date.

Round 2

Reviewer 1 Report

The authors answered all the questions. To the question, "Please specify what the background colour means," the authors responded, "we are sorry we cannot get your point." But they added the information "The shading indicates a dark period" to the figure's legend. That is exactly what I meant.